# Combining MOE Bioinformatics Analysis and In Vitro Pseudovirus Neutralization Assays to Predict the Neutralizing Ability of CV30 Monoclonal Antibody on SARS-CoV-2 Variants

**DOI:** 10.3390/v15071565

**Published:** 2023-07-17

**Authors:** Yajuan Zhu, Husheng Xiong, Shuang Liu, Dawei Wu, Xiaomin Zhang, Xiaolu Shi, Jing Qu, Long Chen, Zheng Liu, Bo Peng, Dingmei Zhang

**Affiliations:** 1School of Public Health, Sun Yat-Sen University, Guangzhou 510080, China; zhuyj58@mail2.sysu.edu.cn (Y.Z.); xionghsh5@mail2.sysu.edu.cn (H.X.); liush333@mail2.sysu.edu.cn (S.L.); wudawei@mail.sysu.edu.cn (D.W.); 2Department of Microbiology Laboratory, Shenzhen Center for Disease Control and Prevention, Shenzhen 518055, China; zhangxiaomin@wjw.sz.gov.cn (X.Z.); shixiaolu831@163.com (X.S.); qujing861105@163.com (J.Q.); chen_l_2011@163.com (L.C.); 3Kobilka Institute of Innovative Drug Discovery, School of Medicine, Chinese University of Hong Kong, Shenzhen 518172, China; liuzheng@cuhk.edu.cn; 4NMPA Key Laboratory for Quality Monitoring and Evaluation of Vaccines and Biological Products, Guangzhou 510080, China

**Keywords:** SARS-CoV-2, mutation, prediction

## Abstract

Combining bioinformatics and in vitro cytology assays, a predictive method was established to quickly evaluate the protective effect of immunity acquired through SARS-CoV-2 infection against variants. Bioinformatics software was first used to predict the changes in the affinity of variant antigens to the CV30 monoclonal antibody by integrating bioinformatics and cytology assays. Then, the ability of the antibody to neutralize the variant antigen was further verified, and the ability of the CV30 to neutralize the new variant strain was predicted through pseudovirus neutralization experiments. The current study has demonstrated that when the Molecular Operating Environment (MOE) predicts |ΔBFE| ≤ 3.0003, it suggests that the CV30 monoclonal antibody exhibits some affinity toward the variant strain and can potentially neutralize it. However, if |ΔBFE| ≥ 4.1539, the CV30 monoclonal antibody does not display any affinity for the variant strain and cannot neutralize it. In contrast, if 3.0003 < |ΔBFE| < 4.1539, it is necessary to conduct a series of neutralization tests promptly with the CV30 monoclonal antibody and the variant pseudovirus to obtain results and supplement the existing method, which is faster than the typical procedures. This approach allows for a rapid assessment of the protective efficacy of natural immunity gained through SARS-CoV-2 infection against variants.

## 1. Introduction

Severe Acute Respiratory Syndrome Coronavirus-2 (SARS-CoV-2) is a highly contagious and dangerous virus that first emerged in late 2019. It has caused a global pandemic of acute respiratory illness, known as Coronavirus Disease 2019 (COVID-19), which poses a significant threat to human life, health, and public safety. Frequent and rapid mutation of the virus has been one of the most notable characteristics of the pandemic, thereby resulting in the emergence of several variant strains. The Omicron variant has emerged as a formidable adversary in the battle against COVID-19 [1,2,3], with multiple subvariants surfacing since its initial appearance. At the end of 2022, a new Omicron subvariant (XBB.1.5) has risen to prominence as the most contagious strain of the virus, quickly spreading worldwide. What is the herd immunity of the entire population to the new variant strain? Do we need to develop a new vaccine for the new strain? These are the questions that we need to consider and address in the long term as we prepare to coexist with SARS-CoV-2.

A disease epidemic is necessary to conduct traditional vaccine efficacy studies, which is time-consuming that provides delayed evaluation results. This delay hinders the development of timely prevention and control measures for the epidemic. In addition, there are various biases in this type of observational evaluation. Comparing the results of individuals who have received the vaccine and those who have not, there may be differences in factors that affect their infection or mortality risks, which may affect the infection rates of the two groups. These differences may confound the evaluation of the vaccine’s efficacy in preventing the disease [4]. Moreover, as vaccination becomes widespread, the number of unvaccinated individuals decreases, thereby making it difficult to obtain data on the incidence and differences in disease rates between vaccinated and non-vaccinated populations, as well as making it difficult to evaluate vaccine effectiveness through traditional methods. Therefore, the current vaccine evaluation process cannot promptly demonstrate the efficacy of vaccines against new variant strains, thereby significantly delaying evaluation results.

However, not all variants have serious and devastating effects. Thus, precisely detecting the efficacy of vaccines against every emerging variant not only consumes significant human, material, and financial resources but also presents remarkable difficulties to the current large-scale rapid analysis capabilities. Therefore, monitoring SARS-CoV-2 variants and determining the efficacy of vaccines against these variants, as well as predicting potential variants that may lead to severe and destructive effects, can guide the development, prevention, and control of COVID-19 vaccines. The most important step in the process of SARS-CoV-2 infection is the binding of the S protein and the organism’s cellular receptor. The neutralizing antibodies produced by the infected virus or vaccination are used to prevent the S protein from binding to the receptor. Therefore, the viral protein’s ability to bind to the antibody can reflect the effectiveness of the antibody. Thus, developing dependable computational methods to predict the impact of missense mutations on antigen–antibody binding is a top priority in current research. Numerous scholars have proposed various methods to forecast the changes in binding free energy (BFE) resulting from mutations, including the TopNetTree model [5], MutaBind2 method [6], SAAMBE-SEQ method [7], SAAMBE-3D method [8], and so on. A study [9] used a bioinformatics molecular docking approach to predict the impact of the angiotensin-converting enzyme 2 (ACE2) receptor when interacting with the receptor binding domain (RBD) of five new coronavirus variants (Alpha, Beta, Gamma, Delta, and Omicron). The results showed that these variants can alter the interaction of S and human ACE2 proteins, lose or create new interprotein contacts, enhance viral fitness by increasing binding affinity, and increase infectivity, virulence, and transmissibility. These methods mentioned above that are highly technical are web pages or codes designed by the original creators, and the public can only upload their own work waiting for feedback on the results. In contrast, the Molecular Operating Environment (MOE) used in this article is one of the bioinformatics software, which is recognized by many of the world’s leading academic journals. The public can operate in the software itself, which is more convenient.

The RBD region of the SARS-CoV-2 S protein is the key site to mediate the virus binding to the ACE2 receptor. In addition, the RBD region is the critical target for the action of neutralizing antibodies and vaccines [10]. Nevertheless, if the RBD key antigenic alteration can be effectively monitored, the early response to the immune escape it caused can be implemented, which is of significant scientific importance. Based on the literature, for most studied antibodies, their epitopes overlap with the ACE2 interface. There is a positive correlation trend between the binding free energy changes of RBD-ACE2 complex mutations and those of all studied RBD-antibody complexes. Moreover, during the evolution of COVID-19, the virus tends to prioritize mutations that promote antigen escape without affecting the binding to the ACE2 receptor protein [11]. Hence, it is more meaningful to investigate the binding free energy changes of RBD-antibody complexes caused by viral mutations. After a rigorous screening process, the CV30 monoclonal antibody was chosen as a reference because of its 3D structure being available in the Protein Data Bank (PDB) protein database and its commercial availability for subsequent testing. This potent neutralizing monoclonal antibody was isolated from a patient with early SARS-CoV-2 infection. The structure of the CV30 complexed with the SARS-CoV-2 RBD reveals that it binds almost exclusively to the concave ACE2 binding epitope of the RBD, also known as the receptor binding motif (RBM). A total of 29 residues of the RBD interacted with CV30, with 19, 7, and 3 residues interacting with the CV30 heavy chain, CV30 light chain, and both the CV30 heavy and light chains, respectively [12].

If it is possible to analyze virus mutations from a structural perspective through bioinformatics and achieve rapid prediction of neutralization capacity in variant strains, it would be a very meaningful work. In the present study, bioinformatics software was first used to predict the changes in the affinity of variant antigens to the CV30 monoclonal antibody by integrating bioinformatics and cytology assays. Then, the ability of the antibody to neutralize the variant antigen was further verified, and the ability of the CV30 to neutralize the new variant strain was predicted through pseudovirus neutralization experiments. Combining MOE bioinformatics analysis and pseudovirus neutralization assay could be able to quickly evaluate the protective effect of immunity acquired through SARS-CoV-2 infection against variants. If a new variant emerges, the neutralizing ability of the immune serum against the new variant can be predicted directly using bioinformatics by calculating the value of the BFE change for the binding reaction between the monoclonal antibody and the RBD of the new variant.

## 2. Materials and Methods

### 2.1. Protein Structure Preparation and Processing

The 3D structures of the CV30 monoclonal antibody and SARS-CoV-2 RBD complex were downloaded from the protein data bank (PDB ID 6XE1). The structure was solved by the X-ray crystal diffraction method at the resolution of 2.75 Å. However, the protein structure downloaded from the internet has to do pre-processing (structure cleaning, correction, hydrogenation, energy minimization, etc.) in the MOE software first. Structure pre-processing was performed via clicking on Compute|Prepare|Structure Preparation in the menu bar of the MOE software. In the pop-up screen, Correct and Protonate3D were selected in sequence. Then, Protein|Design|Residue Scan in the menu bar of the MOE software was clicked on to set up the required variation-related information, and finally, the OK button was clicked to wait for the results. The 3D structure display and analysis were performed by Chimera UCSF [13].

### 2.2. Pseudovirus Neutralization Assay (pVNT)

The cell line used was HEK293T cells with stable high expression of ACE2. Six pseudoviruses were included in the trial, namely SARS-CoV-2 wild type (Genomeditech (Shanghai, China), GM-0220PV07), Alpha (Genomeditech, GM-0220PV33), Beta (Genomeditech, GM-0220PV32), Gamma (Genomeditech, GM-0220PV47), Delta (Genomeditech, GM-0220PV45), and Omicron BA.1 (Genomeditech, GM-0220PV84).

The HEK293T-ACE2 cells were inoculated in 96-well cell culture plates and incubated overnight in a constant temperature incubator containing 5% CO_2_ at 37 °C. Serum samples and pseudovirus were diluted with DMEM complete medium. The serially diluted serum samples were mixed with the diluted pseudovirus solution in equal volumes, and the mixture was incubated at room temperature for 1 h. Subsequently, the 96-well plates with HEK293T-ACE2 cells were removed from the incubator, and the upper layer of the culture medium was aspirated. The diluted pseudovirus serum mixture was then aspirated and added to the 96-well plate lined with HEK293T-ACE2 cells and incubated at 37 °C and 5% CO_2_ for 48 h. A bottle of Bio-Lite Luciferase Assay buffer was added to the Bio-Lite Luciferase Assay substrate, and the mixture was equilibrated to room temperature. Afterward, the plate was removed to be tested from the incubator and equilibrated to room temperature. Then, the supernatant was carefully aspirated and discarded from the well plate. Bio-Lite luciferase assay reagent was added after equilibration at room temperature and repeatedly blown and aspirated 3–5 times to lyse the cells sufficiently. Then, the liquid was transferred to the corresponding 96-well white opaque chemiluminescence assay plate. Chemiluminescence detection was performed using a multifunctional enzyme marker with a chemiluminescence detection function, and the relative light unit (RLU) value was read. Generally, the highest luminescence value was obtained for the wells with only cells and pseudovirus without serum. The luminescence value of the sample wells was inversely proportional to the concentration of neutralizing antibody in the serum, and the lower the luminescence value of the sample wells, the stronger the ability of the sample to neutralize pseudovirus. The pseudovirus infection inhibition rate was calculated based on the following formula.
$$\mathrm{Inhibition}\;\left(\%\right)=(1-\frac{\mathrm{Sample}\;\mathrm{RLU}-\mathrm{Blank}}{\mathrm{Pseudovirus}\;\mathrm{RLU}-\mathrm{Blank}})\times 100\%$$

### 2.3. Calculation of Half-Maximal Inhibitory Concentration (IC50)

GraphPad Prism software was used to calculate IC50 values. First, the GraphPad Prism software was opened, and the XY option was selected. The first item “enter and plot a single y value for each point” was selected for the y value, and “create” was clicked. Then, the antibody concentration and inhibition rate were entered in the X and Y columns, respectively, and “Transform concentration (X)” was selected in the options. Subsequently, “Transform to logarithms” was selected in the pop-up screen. Then, the Analyze interface was entered; “Log (inhibitor) vs. Variable slope (four parameters)” was selected; and finally, “OK” was clicked to get the IC50 value.

## 3. Results

### 3.1. Molecular Operating Environment Predicts Variation in Binding Free Energy Due to Mutations

The 3D structure of the CV30 monoclonal antibody and SARS-CoV-2 RBD complex was obtained by X-ray single crystal diffraction method and was uploaded to the PDB Protein Data Bank on 1 July 2020. This crystal structure’s resolution was 2.75 Å. The spatial binding map of SARS-CoV-2 RBD, ACE2 receptor, and CV30 monoclonal antibody complex was demonstrated using MOE software (Figure 1). As shown, the binding site of the RBD-ACE2-CV30 complex is spatially conflicting with the RBD-ACE2 complex. Theoretically, CV30 monoclonal antibody could prevent the binding of the RBD and ACE2 receptor. Mutant sites of variants in the RBD region were shown in Figure 2 and Table 1. The changes in binding free energy caused by variants in the RBD region of the five variants (Alpha, Beta, Gamma, Delta, and Omicron BA.1) operated by MOE molecular manipulation environment were shown in Table 1. In order to rationalize and verify the results of BFE analysis of MOE, we investigated the interacting residue change on the CV30 and RBD complex structures by displaying the structure using Chimera UCSF software (Figure 2). We carefully checked the interacting detail with CV30 for the mutant residue 417, which occurred beta and gamma. The distance between RBD K417 Lysine and CV30 Y52 tyrosine is less than 3.0 angstrom, indicating a cation–π interaction between the two residues. The mutant of K417T and K417N changes the lysine into an uncharged amino acid (K417T-gamma and K417N-Beta) which will break the interaction to reduce the affinity of antibody to RBD domain. This observation is also consistent with what we observed from the MOE BFE analysis.

### 3.2. Pseudovirus Neutralization Test Results

The pseudovirus carries both green fluorescent protein (GFP) and luciferase reporter genes, and the activity of the pseudovirus-infected cells can be evaluated by observing fluorescence and detecting luciferase activity. The inverted microscope of the HEK293T-ACE2 cell line without pseudovirus and the fluorescent image of the pseudovirus successfully infecting the HEK293T-ACE2 cell line were demonstrated in Figure 3. As shown, the fluorescence of the pseudovirus was strong, which indicated that the pseudovirus can highly infect the cells.

RLU was measured and recorded on the zymograph at the end of each test, and the summarized results were shown in Table 2. Then, the infection inhibition rate of the pseudoviruses was recorded and calculated. The results of the IC50 value of the reactions between CV30 monoclonal antibody and five pseudoviruses (wild type, Alpha, Beta, Gamma, and Delta) obtained by the pseudovirus neutralization assay were presented in Figure 4. However, given that the Omicron BA.1 pseudovirus remained not neutralized at the highest dilution concentration of CV30 monoclonal antibody (11,111.11 ng/mL), it was considered that the IC50 value for the reaction between CV30 monoclonal antibody and Omicron BA.1 pseudovirus was more than 11,111.11 ng/mL.

### 3.3. Combining MOE Bioinformatics Analysis and Pseudovirus Neutralization Assay to Quickly Evaluate the Protective Effect of Immunity Acquired through SARS-CoV-2 Infection against Variants

Bioinformatics was used to predict the binding ability of variant antigens to antibodies, and then the binding ability of antibodies to variant antigens was further verified by pseudovirus neutralization assays. Bioinformatics was combined with cytological tests to quickly evaluate the protective effect of immunity acquired through SARS-CoV-2 infection against variants.

The MOE software interprets the BFE to indicate the binding free energy change, with a more negative value indicating a stronger affinity and a more positive value indicating a weaker affinity. From the neutralization results of six pseudoviruses (wild type, Alpha, Beta, Gamma, Delta, and Omicron BA.1) that have been conducted (Figure 5), the positive and negative signs of the BFE change values predicted by the MOE software may have a weaker sense of guidance in the present research, while the size of |ΔBFE| has a certain predictive effect. From the available results, if the MOE predicted |ΔBFE| ≤ 3.0003, it indicates that CV30 monoclonal antibody has some affinity for the variant strain and can neutralize the variant strain; however, if |ΔBFE| ≥ 4.1539, CV30 monoclonal antibody has no affinity for the variant strain and cannot neutralize the variant. and if 3.0003 < |ΔBFE| < 4.1539, it is necessary to perform a set of neutralization tests with CV30 monoclonal antibody and the variant pseudovirus quickly to obtain results and supplement the existing method, which is faster than the normal situation of organizing. The value of 3.0003 was the largest |ΔBFE| among the several variants for which CV30 still has some neutralizing power. And 4.1539 was the |ΔBFE| corresponding to the Omicron variant for which CV30 has no neutralizing power. Validation with Omicron subvariants BA.2, BA.3, BA.4, and BA.5 showed that the |ΔBFE| of these variants was greater than 4.1539, thereby indicating that the CV30 monoclonal antibody had no affinity for these variants and did not neutralize the variant, and this prediction was consistent with the existing findings. Therefore, the size of |ΔBFE| has a certain predictive effect.

Thus, bioinformatics can be used to directly predict the ability of the immune serum to neutralize the new variant if it emerges. Given that almost everyone is currently infected with Omicron, if we could obtain the structural information and commercial availability for subsequent experiments of the antibody produced after human infection with Omicron, we could replace the CV30 antibody with this antibody.

## 4. Discussion

The emergence of highly transmissible SARS-CoV-2 variants is a great public concern in the current epidemic situation. Thus, monitoring SARS-CoV-2 variants and determining the impact of these mutations and further predicting in advance mutations that could lead to immune escape phenomenon are the current focus and difficulty in the fight against COVID-19. Therefore, it is greatly important to predict the altered affinity of variant antigens to CV30 antibodies using bioinformatics software, further verify the neutralization ability of antibodies to variant antigens by pseudovirus neutralization tests, and then predict and evaluate the protective effect of immunity acquired through SARS-CoV-2 infection against variants. In addition, the CV30 antibody can be replaced with the antibody produced after vaccination. The construction of a predictive evaluation method for the neutralization ability of antibodies against variant strains can help provide preliminary insight into the effectiveness of vaccines at the early stage or even before the emergence of mutations, which is important for the timely adjustment of vaccination strategies and determination of the need to develop new anti-variant vaccines.

Currently, no studies have used the link between the BFE and neutralizing ability to establish a predictive method; however, numerous studies use the BFE obtained from bioinformatics to predict the possible effects of variation. A study [14] obtained the human monoclonal antibody CT-P59 targeting the SARS-CoV-2 S protein RBD from peripheral blood mononuclear cells of patients recovering from neo-coronavirus pneumonia and showed that CT-P59 completely inhibited the binding of RBD to the ACE2 receptor protein by a competitive binding assay using biofilm layer interference techniques. Moreover, the complex crystal structure of CT-P59 and RBD directly observed that CT-P59 blocked the interaction region of the ACE2 receptor with RBD. Jiahui Chen et al. [15] applied the TopNetTree model to predict changes in the binding free energy of the RBD region of the mutated SARS-CoV-2 S protein and the antibody (or ACE2) complex. They initially retrieved more than 200,000 complete SARS-CoV-2 genome sequences from the GISAID database and created a real-time interactive SARS-CoV-2 variation tracker. Detailed variation information can be seen in the variation tracker. Then, a library containing 56 antibody structures was constructed using the antibody structures published on the PDB website. Their 3D features were also analyzed to see directly spatially whether or not the binding sites of S protein and antibody conflict with the binding sites of S protein and ACE2. If, in terms of 3D structures, their binding sites overlap, this indicated that the antibody competes directly with ACE2 for binding to the S protein. Theoretically, this direct competition reduced the rate of viral infection, and such antibodies with high binding capacity would directly neutralize SARS-CoV-2. Then, the authors predicted the binding free energy changes of the variant S protein and ACE2 complex and the variant S protein and antibody by the TopNetTree model. A study [16] also performed a comprehensive computer analysis of the variants of the RBD region to investigate whether or not these variants affect the interaction of the virus with the ACE2 receptor protein. First, they retrieved highly prevalent variants from the GISAID database; then, their structural models were constructed using the SWISS-Model server. Subsequently, the stabilization effect of each variant was evaluated by DUET and DeepDGG software. Finally, molecular docking was applied using Z-Dock and Haddock software. The results revealed that a portion of the variants (e.g., V445E, V445L, and A455V) exhibited higher binding free energy than the wild-type S proteins, and thus these variants had destabilizing effects on binding to ACE2 receptor proteins. Some other variants (e.g., N501Y, K444R, Q493R, and Y505W) displayed lower binding free energies, thereby indicating that they bound stably to ACE2 receptor proteins. Marine E. Bozdaganyan [11] indicated that for most of the antibodies studied, their epitopes overlap with the ACE2 interface. A positive correlation trend can be observed between the value of the binding free energy change of the RBD-ACE2 complex variant and the value of the binding free energy change of the same variant in all studied RBD-antibody complexes. Moreover, during the evolution of SARS-CoV-2, the virus preferentially selected variants that promote antigen escape while not affecting the binding to the ACE2 receptor protein.

This study was the first to combine MOE bioinformatics analysis and pseudovirus neutralization assay to predict the neutralizing ability with CV30 monoclonal antibody as a reference. Thus, it is of more interest to study the changes in binding free energy of RBD-antibody complexes caused by viral mutations. CV30 binds almost exclusively to the concave ACE2 binding epitope of the RBD, also known as the RBM. The RBD region of the SARS-CoV-2 S protein is the critical site that mediates the binding of the virus to the host cell ACE2 receptor and the key target to neutralize antibody and vaccine action. Therefore, it is of great scientific importance to target the effective monitoring of key antigenic alterations in the RBD and the early response to the immune escape it caused. The present study revealed that if the MOE predicted |ΔBFE| ≤ 3.0003, it indicated that CV30 monoclonal antibody had some affinity for the variant strain and can neutralize the variant strain; however, if |ΔBFE| ≥ 4.1539, CV30 monoclonal antibody had no affinity for the variant strain and did not neutralize the variant, and if 3.0003 < |ΔBFE| < 4.1539, it was necessary to perform a set of neutralization tests with CV30 monoclonal antibody and the variant pseudovirus quickly to obtain results and supplement the existing method, which was faster than the normal situation of organizing. This method can quickly predict and evaluate the protective effect of immunity acquired through SARS-CoV-2 infection against variants. If a new variant emerges, the neutralizing ability of the immune serum against the new variant can be predicted directly using bioinformatics. In the current context where almost everyone has been infected with Omicron, the CV30 antibody could be replaced with the antibody produced by the human body after infection with Omicron if we could obtain the structural information and commercial availability of this antibody for subsequent experiments.

However, there were some limitations in this study. No in vivo assays were performed. And only one antibody (CV30 monoclonal antibody) was selected as a reference during the combination of bioinformatics and in vitro cytology assays. Moreover, the number of pseudovirus strains used was limited. The prediction results would be more reliable if other additional antibodies and more pseudovirus strains could be added to complete the method.

## 5. Conclusions

Monitoring SARS-CoV-2 variants and determining the impact of these mutations and further predicting in advance mutations that could lead to immune escape phenomenon are the current focus and difficulty in the fight against COVID-19. This method can quickly predict and evaluate the protective effect of immunity acquired through SARS-CoV-2 infection against variants. Thus, bioinformatics can be used by calculating the value of the BFE change for the binding reaction between the monoclonal antibody and the RBD of the new variant to directly predict the ability of the immune serum to neutralize the new variant if it emerges. Moreover, given that almost everyone is currently infected with Omicron, if we could obtain the structural information and commercial availability for subsequent experiments of the antibody produced after human infection with Omicron, we could replace the CV30 antibody with this antibody. Further studies supplemented with more antibodies and pseudoviruses deserve to be followed.

## Figures and Tables

**Figure 1 viruses-15-01565-f001:**
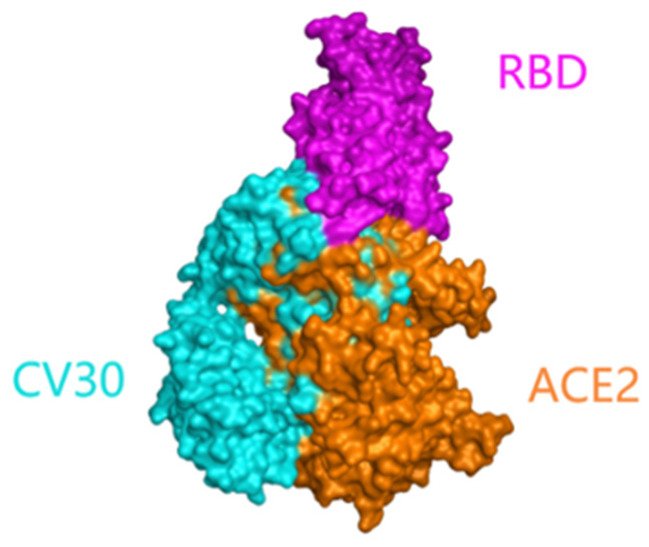
Spatial binding diagram of SARS-CoV-2 RBD, ACE2 receptor, and CV30 monoclonal antibody complex.

**Figure 2 viruses-15-01565-f002:**
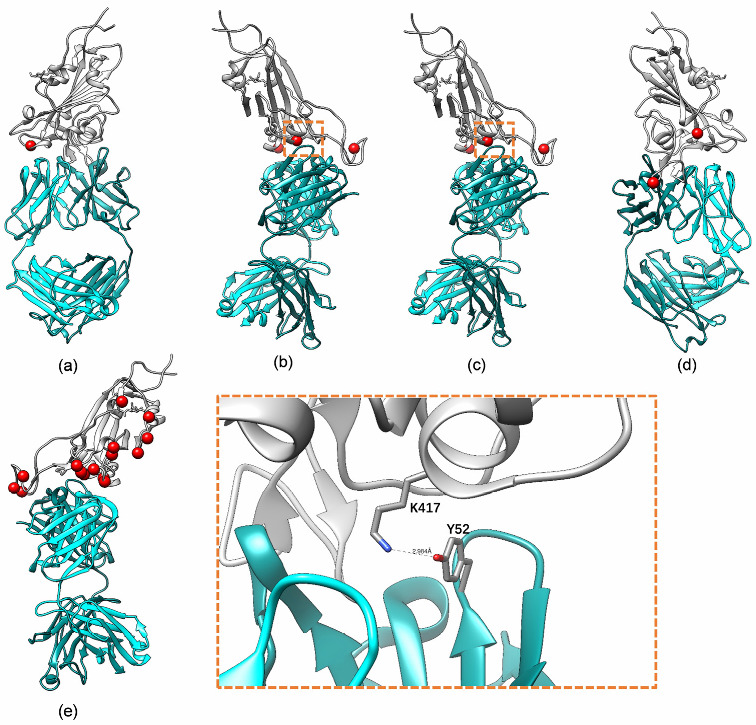
The interacting residue change on the CV30 and RBD complex structures. The grey field referred to RBD, the blue field referred to CV30, and red dots referred to mutation sites in RBD region. (**a**) Alpha, (**b**) Beta, (**c**) Gamma, (**d**) Delta, and (**e**) Omicron BA.1.

**Figure 3 viruses-15-01565-f003:**
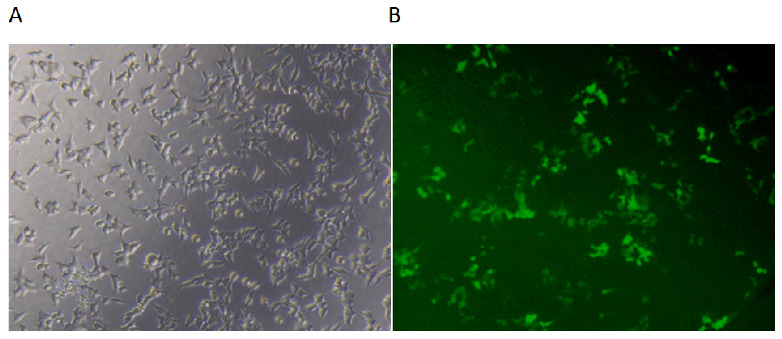
Inverted microscope of HEK293T-ACE2 cell line without the addition of pseudovirus (10 × 20) (**A**) and the fluorescent image of the pseudovirus successfully infecting HEK293T-ACE2 cell line (10 × 20) (**B**).

**Figure 4 viruses-15-01565-f004:**
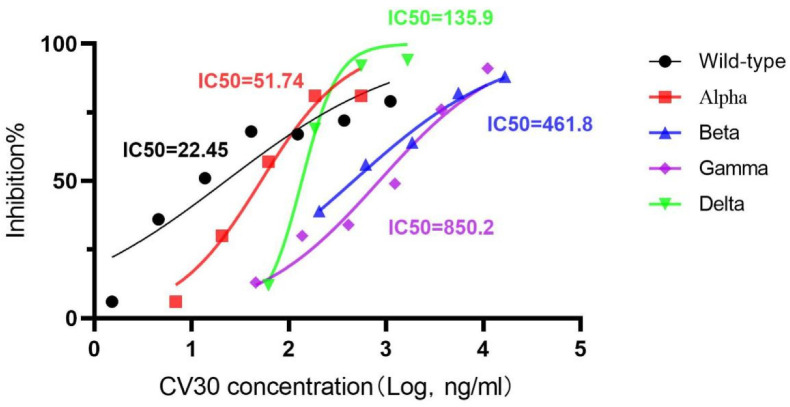
Results of pseudovirus neutralization test.

**Figure 5 viruses-15-01565-f005:**
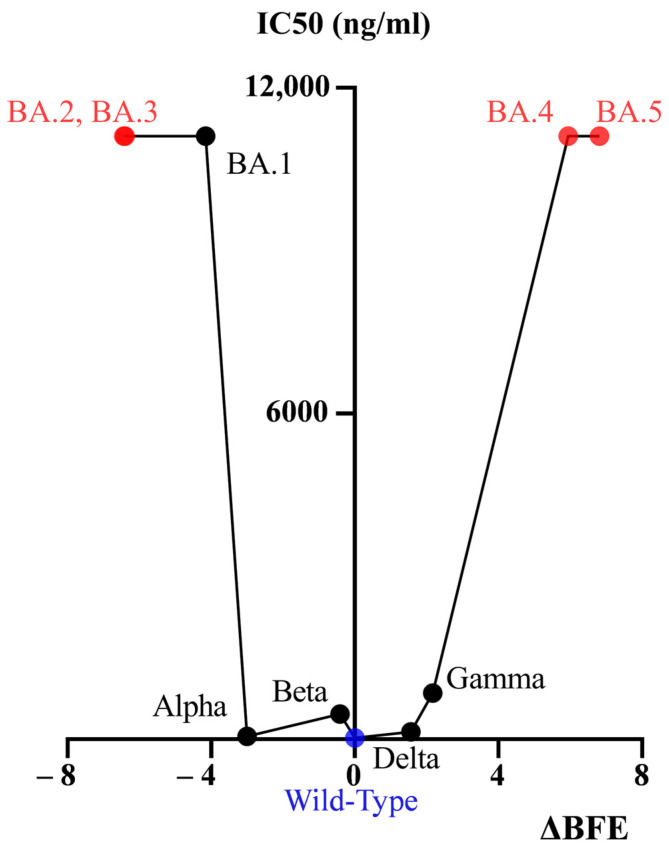
Diagram of BFE change value in relation to IC50 value.

**Table 1 viruses-15-01565-t001:** Changes in binding free energy due to MOE prediction variants.

Pseudovirus	RBD Region Mutation Sites	BFE Change Values (Value of Change in Binding Free Energy Due to Variation in RBD Region Compared to Wild Type)
Wild-type	—	0
Alpha	N501Y	−3.0003
Beta	K417N, E484K, N501Y	−0.4156
Gamma	K417T, E484K, N501Y	2.168
Delta	L452R, T478K	1.5593
Omicron BA.1	G339D, S371L, S373P, S375F, K417N, N440K, G446S, S477N, T478K, E484A, Q493K, G496S, Q498R, N501Y, Y505H	−4.1539

**Table 2 viruses-15-01565-t002:** RLU results of pseudovirus neutralization assay.

Pseudovirus	RLU for Cell Blank Control Wells	RLU for Pseudovirus Control Wells	RLU for Wells with the Addition of CV30	Dilution Concentration of CV30 (ng/mL)
Wild type	332	156,276	34,916	1111.11
	348	170,939	46,380	370.37
	-	-	53,666	123.46
	-	-	52,745	41.15
	-	-	80,319	13.72
	-	-	104,717	4.57
	-	-	153,279	1.52
Alpha	181	140,322	25,055	555.56
	220	121,142	25,595	185.19
	-	-	56,050	61.73
	-	-	91,128	20.58
	-	-	122,564	6.86
Beta	24	12,496	1424	16,666.67
	32	11,097	2157	5555.56
	-	-	4308	1851.85
	-	-	5159	617.28
	-	-	7168	205.76
Gamma	375	240,420	21,549	11,111.11
	480	224,825	55,199	3703.70
	-	-	117,904	1234.57
	-	-	154,378	411.52
	-	-	163,096	137.17
	-	-	201,732	45.72
Delta	181	43041	2711	1666.67
	220	46,267	3651	555.56
	-	-	13,825	185.19
	-	-	39,368	61.73
Omicron BA.1	375	151,281	132,780	11,111.11
	480	152,787	122,761	3703.70
	-	-	119,293	1234.57
	-	-	113,031	411.52

Notes: In general, the highest luminescence values were obtained for wells with only cells and pseudoviruses without the addition of the CV30 monoclonal antibody. And the luminescence values of wells with the addition of the CV30 were inversely proportional to the concentration of the CV30. The lower the RLU of the well with the CV30 indicated the greater the ability of the CV30 at that concentration to neutralize the pseudovirus.

## Data Availability

The data presented in this study are available on request from the corresponding author. The data are not publicly available due to the privacy reasons.

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
