# Peer review of "Combining MOE Bioinformatics Analysis and In Vitro Pseudovirus Neutralization Assays to Predict the Neutralizing Ability of CV30 Monoclonal Antibody on SARS-CoV-2 Variants"

_viruses, 2023, doi:10.3390/v15071565_

Round 1
Reviewer 1 Report
Line 190: Do you mean luciferase reporter genes (typo)?
Line 212: Was there any statistical significance between each variant?
In lines 325-329, another limitation would be no in vivo assays were performed.
It would be great if you could describe how these findings can be used for future studies.
Only minor editing is required.
Reviewer 2 Report
The authors utilize MOE software to simulate various variants of the RBD region of the SARS-COV-2 S protein and make predictions about their affinity for the CV30 monoclonal antibody. Subsequently, they assess the binding capacity of CV30 to different RBD variants through a pseudo-virus neutralization assay (pVNA). The authors emphasize that they develop a predictive method for evaluating CV30's ability to neutralize various SARS-COV-2 variants. However, it should be noted that both MOE and pVNA are commonly employed approaches for studying protein interactions, and they were not exclusively established by this particular manuscript. Moreover, given that the binding affinity between CV30 and different RBD variants determines their interaction, a thorough and meticulous analysis of this interaction should be conducted. Such an analysis should clarify the disparities in binding affinities between CV30 and the various RBD variants. Finally, employing the absolute value of delta BFE (binding free energy) to determine whether CV30 can neutralize the interaction between the S protein and ACE2 is inappropriate because the software used to calculate the binding free energy might not be entirely accurate.
Reviewer 3 Report
1. Authors mentioned the methods like TopNetTree model, MutaBind2 method, SAAMBE
1.SEQ method, SAAMBE-3D etc. were used to forecast the changes in binding free energy resulting from mutations, as well as a bioinformatics molecular docking approach to predict the impact of the angiotensin-converting enzyme 2 receptor when interacting with the receptor binding domain of five new coronavirus variants. Please add more description for these methods regarding their advantages and disadvantages comparing to authors’ method.
2. In the method section, authors mentioned after downloading the 3D structure of CV30 monoclonal antibody, pre-processing including structure cleaning, correction, hydrogenation, energy minimization, etc. was needed in the MOE software. Please briefly describe how to do these steps in MOE software, like the key steps/parameters.
3. Please add how to calculate the BFE change values in the method section.
4. How many biological replicates for each pseudovirus did authors do for BFE calculation?
5. Please combine Figure 3 and Figure 4 as a direct comparison.
6. Please add results description for Table 2.
7. How was |ΔBFE| calculated in the MOE software?
8. Please describe how to determine the |ΔBFE| threshold like 3.0003 or 4.1539 to indicate whether the antibody has affinity for the variant strain and can neutralize the variant strain or not.
9. How to validate the prediction for authors method? Please add the potential validation methods.
Round 2
Reviewer 2 Report
In the previous round of reviewing, I expressed three concerns regarding the main conclusions of this manuscript. Regrettably, none of these concerns have been addressed in the authors' responses. Their explanations lack factual evidence and data support, as a result, appear unsubstantiated. Here, I offer my suggestions to the authors for revising their manuscript in order to strengthen the conclusions of this manuscript.
1. My first concern was the emphasis in the title “establishing a predictive method for neutralizing ability based on the CV30 monoclonal antibody”. As I previously pointed out, none of the methods employed in the manuscript were developed or exclusively established by the authors. In response, the authors argued that jointly using approaches (i.e. molecular docking analysis and pseudovirus neutralization assay) is innovative. However, this argument lacks substantiation and fails to provide any factual basis, rendering it unacceptable. It is imperative for the authors to modify the title and revise the associated statements in the manuscript to accurately convey information to the readers.
2. The second concern was the differences observed in the interactions between CV30 and various RBD variants. Although the authors utilized software to estimate the binding affinity, there was a lack of analysis regarding the interaction modes between CV30 and each RBD variant, as well as a failure to compare the molecular bound conformations among different molecule pairs. For instance, the importance of factors like complementary shape, electrostatic interactions, hydrogen bonding, van der Waals forces, and hydrophobic interactions was not addressed, nor was there an investigation into whether these factors influenced the binding of CV30 and whether they led to unfavorable binding outcomes. The authors countered this concern by stating that they described and analyzed the molecular binding in Table 1, Table 2, and Figure 4. However, these data and descriptions solely pertain to the estimated binding affinity, which does not elucidate the differences in CV30 binding to different RBD variants.
3. My third concern was the conclusion regarding the use of an absolute ΔBFE (Binding Free Energy) value to determine the neutralization ability of the CV30 monoclonal antibody against the SARS-CoV-2 variant strain. As acknowledged by the authors, the calculated BFE may not be entirely accurate. Therefore, it remains unclear to what extent an absolute ΔBFE value can accurately indicate the ability to neutralize the SARS-CoV-2 variant strain. It is crucial to assess the accuracy of the MOE software's predictions for determining neutralization ability by conducting additional simulations. For instance, it is necessary to explore whether random mutations occurring outside the binding regions of CV30 have no impact on neutralization ability and result in a low binding affinity?
Reviewer 3 Report
The answers are reasonable and the added content makes sense. I've no other concerns.
Author Response
Thank you for your recognition of our work